# Rationale for the Use of Radiation-Activated Mesenchymal Stromal/Stem Cells in Acute Respiratory Distress Syndrome

**DOI:** 10.3390/cells9092015

**Published:** 2020-09-02

**Authors:** Isabel Tovar, Rosa Guerrero, Jesús J. López-Peñalver, José Expósito, José Mariano Ruiz de Almodóvar

**Affiliations:** 1Departamento de Oncología Médica y Radioterapia, Servicio Andaluz de Salud (SAS), Avenida de las Fuerzas Armadas 2, 18014 Granada, Spain; mariai.tovar.sspa@juntadeandalucia.es (I.T.); rosa.guerrero.sspa@juntadeandalucia.es (R.G.); jose.exposito.sspa@juntadeandalucia.es (J.E.); 2Instituto de Investigación Biosanitaria, Ibis Granada, Hospital Universitario Virgen de las Nieves, Avenida de las Fuerzas Armadas 2, 18014 Granada, Spain; 3Unidad de Radiología Experimental, Centro de Investigación Biomédica, Universidad de Granada, PTS Granada, 18016 Granada, Spain; jjpenalver@ugr.es; 4Departamento de Radiología y Medicina Física, Facultad de Medicina, Universidad de Granada, PTS Granada, 18016 Granada, Spain; 5Centro de Investigación Biomédica, Universidad de Granad, PTS Granada, 18016 Granada, Spain

**Keywords:** experimental radiotherapy, radiobiology, mesenchymal stem cells, cell therapy, exosome, annexin A1, acute respiratory-distress syndrome, COVID-19

## Abstract

We have previously shown that the combination of radiotherapy with human umbilical-cord-derived mesenchymal stromal/stem cells (MSCs) cell therapy significantly reduces the size of the xenotumors in mice, both in the directly irradiated tumor and in the distant nonirradiated tumor or its metastasis. We have also shown that exosomes secreted from MSCs preirradiated with 2 Gy are quantitatively, functionally and qualitatively different from the exosomes secreted from nonirradiated mesenchymal cells, and also that proteins, exosomes and microvesicles secreted by MSCs suffer a significant change when the cells are activated or nonactivated, with the amount of protein present in the exosomes of the preirradiated cells being 1.5 times greater compared to those from nonirradiated cells. This finding correlates with a dramatic increase in the antitumor activity of the radiotherapy when is combined with MSCs or with preirradiated mesenchymal stromal/stem cells (MSCs*). After the proteomic analysis of the load of the exosomes released from both irradiated and nonirradiated cells, we conclude that annexin A1 is the most important and significant difference between the exosomes released by the cells in either status. Knowing the role of annexin A1 in the control of hypoxia and inflammation that is characteristic of acute respiratory-distress syndrome (ARDS), we designed a hypothetical therapeutic strategy, based on the transplantation of mesenchymal stromal/stem cells stimulated with radiation, to alleviate the symptoms of patients who, due to pneumonia caused by SARS-CoV-2, require to be admitted to an intensive care unit for patients with life-threatening conditions. With this hypothesis, we seek to improve the patients’ respiratory capacity and increase the expectations of their cure.

## 1. Introduction

The investigation into mesenchymal stromal/stem cells (MSCs) has been of outstanding interest in the recent years [1]. Stromal cells are heterogeneous and contain several populations, including stem cells with different multipotential properties, committed progenitors and differentiated cells [2]. In all our experiments, we have used MSCs obtained from the human umbilical cord perivascular area of Wharton’s jelly [3]. We have described these cells and assessed their phenotype [3], self-renewal potential, contractibility, differentiation [3,4,5], clonogenicity, radiosensitivity [6], secretion [5] and antitumoral activity both in basal conditions and after stimulation with X-rays [7,8].

We have recently shown that the combination of human umbilical-cord-derived MSCs cell therapy plus radiotherapy significantly reduces the size of established tumors in mice, both in the directly irradiated tumor and in the distant nonirradiated tumor [7] or in its metastasis [8]. These results support the hypothesis that human mesenchymal stromal/stem cells are radiosensitizers for local tumor radiotherapy, and simultaneously, they represent an effective tool for amplifying the systemic effects of radiotherapy. These out-of-target radiotherapy effects [9,10,11], promoted by MSCs are, in our view, of major interest [9,12].

We have also proved [7,8], that the preirradiation of MSCs triggers an important cellular change that transforms the MSCs into a source of molecules with very interesting pharmacologic proprieties. Amongst these actively secreted molecules, we have identified TRAIL and Dkk3 with very well-known antitumor activities, and annexin A1, whose activities we have previously reviewed [12] and now update here to include new data that demonstrate its anti-inflammatory and antiviral activity and its role in the regulation of hypoxia.

This secretion activity suggests a mechanistic explanation of how activated cells may positively spread their effect far from the place where they are applied. On this basis, we believe that exosomes, heavily loaded with annexin A1, will be liberated in the lungs after cell therapy with irradiated-MSCs cells, and this action would ameliorate symptoms in patients with sepsis in the lungs and in any other organs affected by septic shock.

A significant number of scientific reports are available demonstrating that gap junction, paracrine pathways and exocrine effects can transmit radiation-induced biological effects far from the place where the radiation is applied. These effects are frequently referred to as radiation-induced out-of-target effects. Multiple molecular signaling mechanisms [13] involving oxidative stress [14,15], kinases, inflammatory molecules [16,17], exosomes [8], microvesicles are postulated to contribute to bystander short- and long-range effects [12]. The anticancer immune response may also be activated by ionizing radiation, and a combination of different treatment strategies is promising in this field [11,18]. The activation of the immune system by the irradiated tumor to trigger the beneficial abscopal effect is decisively improving radiotherapy applications and their outcomes [19,20,21,22].

## 2. Mesenchymal Stromal/Stem Cells and Radiotherapy

It is generally acknowledged that MSCs can be found ubiquitously in many tissues and are not limited to those of mesodermal origin, such as bone marrow, adipose, muscle and bone.

Previous reports suggested a protective role for MSCs when combined with radiotherapy (RT) [23,24]. In effect, due to their properties, MSCs may be recognized as a therapeutic tool for treating radiation-induced tissue damage [25,26,27]. Several reports have shown that MSCs skillfully home onto neoplastics tissues [28,29] and together with tissue recovery functions MSCs prepare the microenvironment by controlling inflammatory processes to reduce the inflammation grade [30,31], where they might have the greatest therapeutic impact in vivo [32]. However, the amount of mesenchymal stromal/stem cells that are up-taken into injured tissues may not be sufficient to explain their strong overall protective effect.

The bioactivation of MSCs may be obtained indifferent ways [33] and the molecules secreted by the activated MSCs (MSCs*) might have an impact on several immune-cell lineages, establishing an advantageous sphere far away from its original location. We have proposed that exosomes liberated from radiation-activated MSCs* perform important intratumoral and systemic actions [8,12].

We are aware that cellular therapy with MSCs can be problematic in cancer therapy [12]. Therefore, it is important to emphasize that following irradiation MSCs become senescent and the senescence is associated with production of a senescent-associated secretory phenotype (SASP). The SASP has antitumor activity since it may induce senescence of neighboring cells by paracrine action [34]. Nevertheless, SASP can modify its composition and become much richer in proinflammatory factors and it has become evident that tumor-associated MSCs have a positive effect on tumor growth and the spread of metastasis [35] through the acquisition of a chemo- and radiotherapy resistance mechanisms [36], and it has been suggested that tumor cells can misuse SASP for their own growth [37]. On the other hand, it has been communicated that, in an inflammatory condition, the exosomes contained in the cancer cell secretome might have a role in the change of the normal MSCs cell phenotype toward a malignant one [36], which could be an impediment to MSCs therapeutic use.

Nevertheless, whether this innate tropism of MSCs toward the tumors and metastatic foci is related with cancer promotion or suppression remains controversial [35,37,38,39,40], and further studies on the interactions between cancerous cells and stromal components of tumor microenvironments have been proposed, which is imperative to allow the progress of more suitable treatments for cancer.

It is generally accepted that MSCs-based therapies are of major importance in regenerative medicine and, perhaps, in the future a solution for many other medical problems. However, the success of MSCs therapy relies on the efficiency of its administration and the biodistribution, engraftment, differentiation and secreting paracrine factors at the target sites [41]. Until now, there has been no universal delivery route for mesenchymal stromal/stem cells (MSCs) for different diseases [42]. In fact, efficient homing and migration toward lesion sites play an important role and the local transplantation of MSCs in spatial proximity to the lesion, as well as the systemic administration routes are being carefully explored [43]. There is growing evidence that mesenchymal stromal/stem cells based immunosuppression was mainly attributed to the effects of MSCs-derived extracellular vesicles [44] although it seems clear that transplanted MSCs can indeed leave the blood flow and transmigrate through the endothelial barrier, and reach the lesion site [45]. So, both mechanisms must be accepted until the underlying processes are better understood.

We know that in an uninjured mouse, exogenous intravenously injected MSCs rapidly accumulate within the lungs and are cleared from this site to other organs, such as the liver, within days [46]. As far as up-take in the lungs is concerned, the MSCs are able to release a wide variety of soluble mediators, including anti-inflammatory cytokines [47], antimicrobial and angiogenic macromolecules, and exosomes and microvesicles that are secreted to extravascular spaces [48]. All of the above leads us to believe that the amount of MSCs cells that engraft onto injured tissues may not be sufficient to account for their robust overall protective effects.

On the other hand, the induction of the mechanisms of epithelial and endothelial wound healing and the angiogenesis promotion has been attributed to macromolecules included in the exosomes released by the MSCs cells, which act as tools for defending the intestines from the damage produced by necrotizing enterocolitis experimentally induced in animal models [29]. This has been highly promising [23,49], and MSCs may be a well-thought-out therapeutic tool to treat radiation-induced tissue damage [30]. It is essential to highlight that the group of Chapel et al. has started a phase 2 clinical trial (ClinicalTrials.gov Identifier: NCT02814864) for the handling of severe collateral healthy tissue damage after radiation therapy in patients with prostate cancer, and this clinical trial is sustained by numerous reports focused on the use of MSCs for improving the damage severity on normal tissues after radiation treatment [46,50,51]. However, the damage severity and the mechanisms involved in the control of side effects after radiotherapy [52], as well as the role of MSCs in healthy-tissue radio-protection, are quite unknown.

We have included in Figure 1 a graphic summary of the widespread actions done by MSCs and MSCs*.

## 3. Radiation-Activated Mesenchymal Stromal/Stem Cells

When we studied the exosome cargo before and after the activation of MSCs with RT, we discovered significant disparities in the results of the proteomic assessment of both samples. We described that there are qualitative, quantitative and functional differences amongst the proteins contained in the exosomes obtained from basal MSCs and activated MSCs* [8]. For more information in [8] see Supplementary Materials, additional file 1.

These findings demonstrate the profound metabolic change that these activated cell exosomes have undergone and the consequences after activation with radiation. Amongst the proteins representatives in exosomes released from MSCs*, we highlight the key components of cell–cell or cell–matrix adhesion and include annexin and integrins [8]. Between them, the presence of annexin A1 (ANXA1) is very noteworthy because it is always present in the exosomes released from MSCs* and constantly absent in MSCs. We verified these results using quantitative mRNA–PCR to measure the mRNA of this protein in MSCs and MSCs* and confirmed that mRNA is spectacularly induced in MSCs after irradiation [8]. After measuring quantitatively the mRNAs of the proteins of TRAIL, Dkk3 and ANXA1 in umbilical cord stromal stem-cells, before and after cell stimulation with 2 Gy low-energy transfer ionizing radiation, our previously published results [8] show a clear increase in their intracellular levels, compared with the levels found in basal situations (see these results in [8] supplementary material, Figure S2) and notice that the levels of mRNA of TRAIL and Dkk3 at 48 are strongly increased in treated cells compared to the basal levels (*p* < 0.001), whereas the levels of mRNA of ANXA1 are strongly increased at 24 h, and dramatically at 48 h of cell treatment, with the statistical differences found 24 and 48 h being very significant (*p* < 0.0001), which supports the massive presence of ANXA1 in the exosomes released by the radiation-stimulated MSCs.

## 4. Annexin A1 in the Inflammation and Hypoxia Processes Control

We stated that the existence of ANXA 1 in the exosomes separated from the culture medium of activated MSCs* and the absence of this protein in the medium withdrawn from the nonirradiated MSCs is a relevant outcome in our previous studies [8].

In relation with this protein, we would like to emphasize that after more than 30 years of research, annexins have been clearly recognized as key elements to control immune responses. The prototype component of this family, ANXA1, has been highly recognized as an anti-inflammatory factor involving cell mobility and the response of several components of the innate immune system [53]. However, it has now been recognized that ANXA1 also has important implications in maintaining homeostasis, fetal development, aging processes and in the evolution of several diseases such as cancer [54,55]. Inflammation is a tightly regulated mechanism, initiated following tissue damage or infection. If unrestrained or unsolved, the inflammation may lead to further tissue damage and give rise to persistent inflammatory diseases and autoimmunity with eventual loss of organ function. It is now evident that the outcome of inflammation is an active process that occurs during an intense inflammatory incident [56]. After MSCs activation, the released ANXA1 might diminish the gathering of neutrophils in the tissue injured in several ways. Additionally, ANXA1 promotes neutrophil apoptosis and acts on macrophages to stimulate the phagocytosis and the removal of dead neutrophils [56,57], and leads to the rapid reconstruction of tissue homeostasis. Inflammation resolve is controlled by several endogenous factors involving macromolecules and proteins, such as ANXA1, and their presence is relevant in many diseases [58]. The study of ANXA1 in relationship with the innate immune system has focused mainly on the anti-inflammatory and proresolving actions through its binding to the formyl-peptide receptor 2 (FPR2)/ALX receptor. There is much evidence that ANXA1, and its mimetic peptides [58], may have an important role in alleviating complications associated with ischemia–reperfusion injury [59]. Moreover, the presence of chronic inflammation in tumors is common and facilitates tumor growth, metastatic dissemination and treatment resistance [60]. Physical abnormality of tumor vasculature, including its chaotic structure, enlarged interstitial pressure, increased stiffness and hypoxia, are physical barriers in tumor treatment [61] are inspiring new anticancer strategies aimed at targeting the tumoral tissue to normalize these physical irregularities [61,62].

ANXA1 is an endogenous inhibitor of NF-κB that can be induced in cancer cells and experimental tumors by potent anti-inflammatory glucocorticoids and modified nonsteroidal anti-inflammatory drugs [49]. In this context, ANXA1 has long been classified as an anti-inflammatory protein due to its actions on leukocyte-mediated immune responses. However, it is now well known that ANXA1 has extensive effects further from the immune system, with consequences in maintaining the homeostatic atmosphere within the whole body due to its capacity to influence cellular signaling, hormonal secretion and diseases [63]. Upon an injury, epithelial wound shutting is a excellently adjusted process that re-establishes homeostasis, but in chronic diseases it is related with nonhealing vascular lesions; in this processes ANXA1 is involved as a preresolving mediator [64].

Moreover, new studies indicating an intracellular function of ANXA1 have now been published. In effect, using AnxA1 knockout mice, it has been noted that ANSA1 is essential for IL-1β release both in vivo as in vitro [65]. Furthermore, we know that ANXA1 colocalize and exactly connect with NLRP3, suggesting the activity of ANXA1 in inflammasome initiation is independent of its anti-inflammatory role via FPR2 [65]. These mechanisms, which could be of major importance in the resolution of lung inflammation and in septic shock through cytokine storm control, deserve more research.

## 5. Annexin A1 in the Treatment of Inflammation

The significance of annexin A1 (ANXA1), a 37 kDa monomeric protein, to stress response is that its synthesis and release are controlled by glucocorticoids (GCs). After release, it has been shown that ANXA1 could strongly downregulate polymorphonuclear leukocyte migration into inflammatory sites and accelerate their apoptosis, upregulating the monocyte migration into the inflammatory sites [66].

Recently, the role of ANXA1 in the treatment of acute radiation-induced lung damage has been studied and the causes of its action examined [67]. Neuroinflammation initiated by damage-associated molecular patterns has been implicated in adverse neurological outcomes following lethal hemorrhagic shock and polytrauma [68]. Results obtained by Ma Q. et al. [68] show that attractive proresolving pharmacological approaches, such as annexin-A1 biomimetic peptides, can efficiently attenuate neuroinflammation and reveal a novel complex role for ANXA1 as a therapeutic and a prophylactic drug due to its ability to strengthen endogenous proresolving, anti-thrombo-inflammatory mechanisms in cerebral ischemia–reperfusion injury. Finally, it has been announced that recombinant human ANXA1 may represent a novel candidate for the treatment of diabetes type 2 and/or its complications [69,70].

## 6. Annexin A1 and Lung Diseases

Endogenous glucocorticoids are proresolving intermediaries, a model of which is the endogenous glucocorticoid-regulated protein annexin A1. Because silicosis is an occupational lung disease typified by persistent inflammation and fibrosis, models regarding this illness have been studied to test the therapeutic properties of the ANXA1 on experimental silicosis [66]. The authors have demonstrated that the therapeutic administration of N-terminal peptide of ANXA1 (Ac2-26) in ischemia–reperfusion-provoked lung injury might substantially attenuate the lung edema and proinflammatory cytokine production, thus reducing oxidative stress, apoptosis, neutrophil infiltration and lung tissue injury, perhaps via the activation of the N-formyl peptide receptor [66].

A similar result was published in an experimental study made with animals affected by bleomycin-induced lung fibrosis that were treated with an ANXA1 peptido-mimetic, administrated prophylactically (from day 0 to 21) or therapeutically (from day 14 onward), which improved signs of both inflammation and fibrosis [71]. Together these data show a pathophysiological relevance for ANXA1 in lung inflammation and in fibrosis, and may open up a new approach for the pharmacological handling of pneumonia and lung fibrosis. Currently, the resolution of inflammation, once considered to be a passive process, has recently been revealed to be an active and precisely controlled process. In the resolution stage of acute inflammation, new mediators, including lipoxins and resolvins, which are members of the specific proresolving mediators of inflammation, are released [72].

Acute lung injury and the more severe forms of acute respiratory distress syndrome, ALI/ARDS, are relatively common syndromes in seriously ill patients and are related with a high rate of morbidity and mortality. Recently, new evidence has shown that the resolution of inflammation might be an active and highly regulated process. Specific proresolving mediators (SPMs), have been proved to produce strong immune-resolving effects, such as cell proliferation, migration and the clearance of apoptotic cells and microorganisms. Therefore, the effective and timely control of inflammation could be the key step to maintain effective host defense and the restoration of homeostasis. Therefore, this reveals a new mechanism for pulmonary edema fluid reabsorption in which SPMs, amongst them annexin A1, might offer new chances to design “reabsorption-targeted” treatments with high levels of precision in controlling acute lung injury [73]. It is also widely acknowledged that to survive, edema fluid should be removed for patients with ALI/ARDS [74].

Moreover, lung endotoxemia is characterized by neutrophil accumulation, enlarged vascular permeability and parenchymal damage. In relation with toxic problems, it has been proposed that the molecular reactions stimulated by ANXA1 peptidomimetic Ac2-26 lead to the control of leukocyte activation/migration and both cytokine production and lung injury that are generated by lipopolysaccharides [75]. It was also published that ANXA1 may accelerate the resolution of inflammation in acute radiation-induced lung damage through the inhibition of IL-6 and myeloperoxidase inflammatory cytokines, demonstrating that ANXA1 may have a therapeutic role as treatment target for acute-radiation lung damage [76].

Moreover, it is well known that pattern recognition receptors (PRRs) are key elements in the innate immune response. FPR2/ALXR, a receptor modulated for specialized proresolving mediators of inflammation, amongst them annexin A1, has been shown to be one of the receptors implicated in inflammation process control. This has encouraged the research community to search for and develop new anti-inflammatory/proresolution small molecules to control inflammation through the activation of FPR2/ALXR [44].

We believe that the protective function of the ANXA1-FPR2 signaling axis recently described in viral infections it is very important [60]. The formyl peptide receptor (FPR) 2 is a pattern recognition receptor that, in addition to proinflammatory, pathogen-derived compounds, also recognizes the anti-inflammatory endogenous ligand annexin A1 (ANXA1), and it has been shown that ANXA1, via FPR2, controls inflammation and bacterial dissemination during pneumococcal pneumonia by promoting host defenses, suggesting ANXA1-based peptides as a novel therapeutic strategy to control pneumococcal pneumonia [77].

In this context, it has been described that mice with the influenza A virus (IAV) infection in the murine model treated with ANXA1 displayed significantly attenuated pathology upon a subsequent IAV infection with significantly improved survival, impaired viral replication in the respiratory tract and less severe lung damage.

## 7. COVID-19: The Magnitude of the Problem

Most countries in the world are suffering a significant spread of SARS-CoV-2, causing pandemic effects. The clinical presentation of the SARS-CoV-2 infection varies from asymptomatic or with light symptoms to clinical situations characterized by respiratory insufficiency requiring mechanical ventilation and intensive care, to multiorgan dysfunction syndrome with signs and symptoms such as sepsis, septic shock and multisystem failure. It also is true, unfortunately, that all the countries in the world do not have the capacity to solve this problem due to the lack of therapeutic measures that could have the appropriate impact. The problem is massive. Therefore, there is a great need to contemplate new methods to improve patients’ biological resistance to SARS-CoV-2 by using mesenchymal stromal/stem cells [78]. We know that SARS-CoV-2 invade cells through the ACE2 receptor widely expressed in human cells, including the alveolar epithelium and the capillary endothelium. The MSCs are ACE2 negative. So, the transplanted cells are unable to participate in the spread of the infection.

For the healthcare services, the two key imperative necessities in the SARS-CoV-2 infection are to hinder and reduce infection rates, and to decrease the death rate of those infected. The accumulating epidemiological analyses, connected with country-based mitigation strategies, and with estimations that about 80% COVID-19 patients have mild or asymptomatic disease, 14% severe disease, and 6% are critically ill, support a permanent need for the treatment of SARS-CoV-2 infection and COVID-19 pneumonia in the long term.

According to preliminary estimates of severity that were based on a recent analysis of data from EU/EEA countries and the UK available in the *European Surveillance System TESSy* and online country reports (for countries whose data were incomplete or missing in *TESSy*) and summarized by the *European Centre for Disease Prevention and Control* (*ECDC*), we know that amongst all the cases of patients affected, hospitalization has occurred in 32% of cases reported from 26 countries, and cases with severe illness (requiring ICU and/or respiratory support) have accounted for 2.4% cases reported from 16 countries. Moreover, amongst hospitalized cases, severe illness was reported in 9.2% of hospitalized cases in 19 countries and death occurred in the 11% of the hospitalized cases in 21 countries. The age-specific hospitalization rates amongst all cases showed elevated risk amongst those aged 60 years and over. Finally, a strong estimate for the COVID-19 case death rate is still lacking and theoretically biased by partial outcome data and differences in testing policies and procedures.

The number of people affected worldwide is progressive and continuously growing, and SARS-CoV-2 has infected more than 24.5 million people and killed more than 830,000 people in different countries, areas or territories with cases (ECDC on 28 August 2020). The worldwide lethality (average) is ≈3.38% with a range of 0.1% to 14.0% depending on the country.

The magnitude of the problem is enormous and terrifying.

## 8. Clinical Trials of MSCs Transplantation in Patients with COVID-19 Pneumonia

MSC products are quickly arising as promising treatment candidates for the COVID-19 pandemic. It is well known that septic shock is associated with a considerable viral load in terms of both mortality and morbidity for survivors of this illness. Preclinical sepsis studies advise that mesenchymal stromal/stem cells (MSCs) may moderate inflammation, improve pathogen clearance and tissue repair and reduce death. Because MSCs have not been assessed in humans with septic shock, a clinical trial that examines safety and tolerability of MSCs is mandatory before proceeding to a randomized controlled trial to study patient outcomes. This has been performed by L.A. McIntyre et al. [79] and their results show that the infusion of freshly cultured allogenic bone-marrow-derived MSCs, up to a dose of 3 million cells/kg, into patients with septic shock seems safe and, consequently, the results of the phase I dose escalation and safety trial provide researchers with the rationale and argument to now conduct larger trials to study the efficacy of MSCs in a clinical trial in patients with septic shock [80]; the clinical trial is registered with the www.clinicaltrials.gov (NCT02421484) reference.

Preclinical and early clinical data suggest that human umbilical cord stromal MCSs, because of their anti-inflammatory and immunomodulatory actions, are able to heal tissues affected and thus improve recovery rates [81]. Additionally, this treatment also seems to be antimicrobial. Two recent studies from China [78,82] have examined whether MSCs could be useful for treating SARS-CoV-2/COVID pneumonia, based on known immune modulatory and reparative abilities of stem cells. Both studies show an outstanding reversal of symptoms, even in severe to critical circumstances. These clinical studies not only recognize a novel therapeutic approach, but also the reality of natural processes able to reduce acute inflammatory pneumonia.

Following the intravenous transplantation of MSCs, a noteworthy population of cells accumulates in the lung, which together with their immunomodulatory effect, could protect alveolar epithelial cells, recover the pulmonary microenvironment, avoid pulmonary fibrosis and cure lung dysfunction. It has been suggested that MSCs have cured or significantly improved the functional outcomes of seven patients without any detected side effects. The pulmonary function and symptoms of these seven patients were significantly improved in two days after MSCs transplantation. Furthermore, the gene expression profile revealed MSCs were ACE2^-^ and TMPRSS2, which showed that the MSCs were free from the SARS-CoV-2 infection. Thus, the intravenous cellular transplantation was safe and efficient for handling in patients with COVID pneumonia, particularly for the patients in a seriously severe condition [78].

Given the uncertainties in this area, Golchin et al. [83] have reviewed published clinical trials and hypotheses to offer useful information to researchers and those involved in stem-cell therapy. In their study, they considered a new approach to enhance patients’ immunological responses to COVID-19 pneumonia using MSCs and debating the aspects of this proposed treatment. However, currently, there are no approved MSC-based approaches for the prevention and/or treatment of COVID-19 patients; nevertheless, clinical trials are ongoing.

The immunomodulatory and anti-inflammatory properties of MSCs in the treatment of respiratory diseases have been confirmed by 17 completed clinical studies, and also more than 70 trials have been registered in this regard (https://clinicaltrials.gov).

Many of the critically ill COVID-19 patients are in a hypercoagulable or procoagulant situation and with a high probability for disseminated intravascular coagulation, thromboembolism and thrombotic multiorgan catastrophe, another cause of the high death rate. Therefore, it is mandatory to only use well-characterized and safe MSCs in the most urgent and experimental treatments [84]. Moreover, in order to alleviate patients with SARS-CoV-2 infection, the obvious risk of adverse thrombotic reactions after the transplant of high doses of poorly typified cell product, an obligatory a set of significant procedures for combining innate immune hemocompatibility examination into the usual patients’ characterization and clinical procedures, before applying MSCs cell therapies has been proposed [84].

Of course, cost effectiveness and the speed of medicinal formulation and transport are topics to be considered for MSCs-based therapy for COVID-19, but without a doubt, whatever the cost the life of a human being is priceless. Nevertheless, the clinical use of MSCs therapy to treat COVID-19 seems promising. Therefore, bearing in mind that MSCs therapy could become an important contribution to terminate the high COVID-19 death rates and prevent long-term functional side effects in those who survive disease, it is essential that the funding agencies invest more into the development of MSCs suitable for safe clinical applications [71].

However, it is very important to underline that scientists are tirelessly trying to obtain a vaccine for SARS-CoV-2 infection and COVID-19 pneumonia, as well as therapeutics to treat this disease [83], and that now a vaccine to protect against SARS-CoV-2 infection has been assessed for safety, tolerability and immunogenicity of a recombinant adenovirus type-5 (Ad5) vectored vaccine expressing the spike glycoprotein of a grave acute respiratory syndrome coronavirus 2 (SARS-CoV-2) variety [85]. These recently published results show that the vaccine is safe and immunogenic at 28 days postvaccination. Humoral responses against SARS-CoV-2 hit the highest point at day 28 postvaccination in healthy adults, and quick specific T-cell responses were observed from day 14 postvaccination. These findings imply that the Ad5 vectored SARS-CoV-2/COVID vaccine deserves more research [85] and an ongoing phase 2 trial in China (NCT04341389) will offer more data on the safety and immunogenicity of the Ad5 vectored SARS-CoV-2/COVID-19 vaccine. The progress in this field is extremely fast, and an excellent update on the subject can be found in [86].

## 9. Conclusions and Perspectives

The present global health crisis involving the appearance and rapid spread of a new coronavirus has encouraged the worldwide scientific community to consider how it can help to combat this mounting viral pandemic.

Amongst all the different mesenchymal stromal/stem cells that might be used, umbilical cord stem cells seem to be the most desirable for a series of reasons that have been very well explained by S. Atluri et al. [81]. Considering together both the previous reports and our own knowledge, and research on the exceptional abilities of proliferation [5,7], secretion [4] and differentiation [17,71] of the umbilical cord mesenchymal stromal/stem cells that we have investigated [7,8], we have also decided to recommend umbilical cord mesenchymal stromal/stem cells as a vehicle for annexin A1 for septic shock treatment.

The activation of these MSCs with a 2 Gy low-LET radiation dose produces an important increase in the cell-released exosomes and these nanovesicles, which can reach all the tissues and organs affected, contain a very specific load of proteins, including annexin A1 [8,12], whose activity in situations of infection, inflammation and hypoxia has been intensively discussed in the previous sections of this paper. This protein together with the endothelium-repair functions characteristic of MSCs must play a major role in the treatment of the septic shock and pneumonia related with SARS-CoV-2 infection.

Moreover, it is generally accepted that the efficacy of transplanted MSCs actually seems to be independent of the physical proximity of the transplanted cells to damaged tissue. Supposedly a vectorized signaling system, we now believe that the exosomes released from radiation-activated-MSCs cells can reach other organs different from the lungs, where they will be up-taken after intravenous injection and thus extend the anti-inflammatory and antimicrobiological effects of the treatment, to cover systemic problems such as the treatment of patients with septic shock in general and for COVID-19 at this particular time.

This hypothesis provides a rationale for the therapeutic efficacy of MSCs and their secreted exosomes in patients with clinical conditions characterized by respiratory failure necessitating mechanical ventilation and medical assistance in the intensive care unit, for multiorgan insufficiency and systemic manifestations such as sepsis, septic shock and multiple organ dysfunction cases.

Lastly, a scheme for our hypothetical cellular therapy in patients with acute respiratory distress syndrome would be an intravenous infusion of 6 million/kg of patient-weight divided into two parts: (a) 3 million nonirradiated-MSCs/kg of patient-weight, to take advantage of the protective, regenerative and repair MSCs-effects at the lung–vasculature and (b) 3 million preirradiated-MSCs*/kg of patient-weight, to achieve, as soon as possible within the patients, the loaded-exosomes with ANXA1 that clinical-grade umbilical cord MSCs* are able to produce after radiation stimulation and thus, take advantage of the extensive range of anti-thrombo-inflammatory, antiviral and immunomodulatory actions associated with this protein.

Finally, we want to clarify that this paper only presents a hypothesis and that the possibility of treating patients is still far off because we lack the necessary experimental data, which would prove the applicability, efficiency and security necessary to further the hypothesis in its transition from the laboratory bench to the patient’s bed. Therefore, more work is necessary to promote this idea and use activated MSCs* as a therapy for patients with COVID-19, but that is our challenge and we are optimistic of a positive outcome.

## Figures and Tables

**Figure 1 cells-09-02015-f001:**
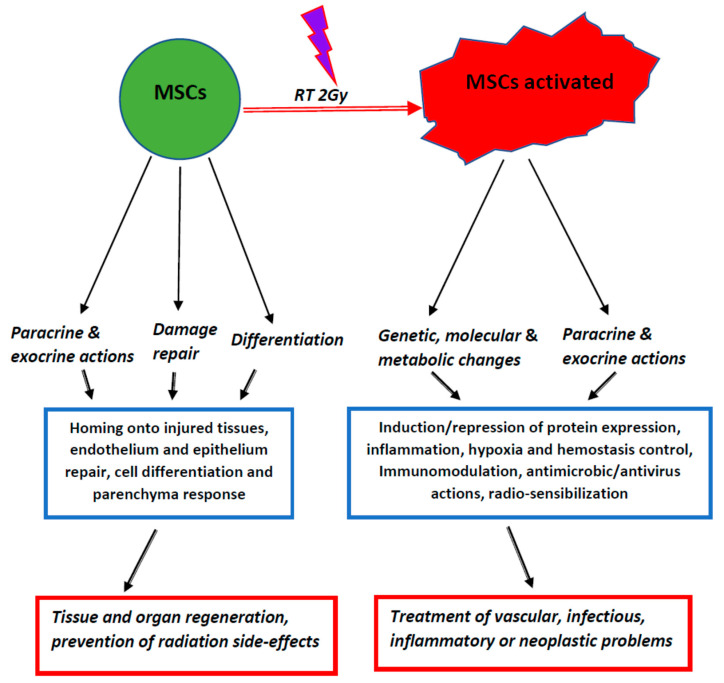
Graphic and schematic summary of cell actions, tissue response and possible therapeutic application of mesenchymal stromal/stem cells (MSCs) and activated MSCs.

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
