# Peer review of "Rationale for the Use of Radiation-Activated Mesenchymal Stromal/Stem Cells in Acute Respiratory Distress Syndrome"

_cells, 2020, doi:10.3390/cells9092015_

Round 1
Reviewer 1 Report
The authors proposed the use of radiation-activated mesenchymal stem cells in acute respiratory distress syndrome.
The review is of interest and also the hypothesis is worthwhile.
There are some issues that have to be addressed before considering it suitable for publication.
The term mesenchymal stem cells is not correct. Authors have to indicate a detailed and proper definition for mesenchymal stromal cells (MSCs) in the “Introduction” paragraph. Stromal cells are heterogeneous and contain several populations, including stem cells. The authors should better explain that the isolation of MSCs, according to current criteria, produces heterogeneous, non-clonal cultures of stromal cells containing stem cells with different multipotential properties, committed progenitors and differentiated cells (see for example PMID: 26423725).
The authors wrote: This finding correlates with a dramatic increase in the anti-tumour activity of the exosomes secreted by pre-irradiated mesenchymal-cells.
The authors did not explain in detail why do irradiated MSCs secrete anti-tumor factors. Indeed, following irradiation MSCs become senescent and the senescence is associated with production of the so senescent associated secretory phenotype (SASP). The SASP has anti-tumor activity since it may induce senescence of neighboring cells by paracrine action. Nevertheless, SASP can modifies its composition and become more and more reach in pro-inflammatory factors. The presence of pro-inflammatory factors in the secretome of irradiated MSCs could be an obstacle to their therapeutic use. Authors should address all these issues also taking into account that irradiated MSCs undergo senescence (see for example PMID: 27288264)
Author Response
Reviewer 1
a) The term mesenchymal stem cells is not correct. Authors have to indicate a detailed and proper definition for mesenchymal stromal cells (MSCs) in the “Introduction” paragraph.
We have rewritten this paragraph that now says:
The investigation into mesenchymal stromal/stem cells (MSCs) has been of outstanding interest in the recent years (1). Stromal cells are heterogeneous and contain several populations, including stem cells with different multipotential properties, committed progenitors and differentiated cells (2). In all our experiments we have used MSCs obtained from the human umbilical cord perivascular area of Wharton's jelly (3). We have described these cells and assessed their phenotype (3), self-renewal potential, contractibility, differentiation (3-5), clonogenicity, radiosensitivity (6), secretion (5) and antitumoral activity both in basal conditions and after stimulation with X-rays (7, 8).
Lines 47-54 in the new manuscript
- b) The authors wrote: This finding correlates with a dramatic increase in the anti-tumour activity of the exosomes secreted by pre-irradiated mesenchymal-cells.
We are in accordance with the reviewer. To amending this paragraph we have changed it and now says:
This finding correlates with a dramatic increase in the anti-tumour activity of the radiotherapy when is combined with MSC or with pre-irradiated mesenchymal-cells.
Lines 31-32 in the new manuscript
- c) The authors did not explain in detail why do irradiated MSCs secrete anti-tumor factors. Indeed, following irradiation MSCs become senescent and the senescence is associated with production of the so senescent associated secretory phenotype (SASP). The SASP has anti-tumor activity since it may induce senescence of neighboring cells by paracrine action. Nevertheless, SASP can modifies its composition and become more and more reach in pro-inflammatory factors. The presence of pro-inflammatory factors in the secretome of irradiated MSCs could be an obstacle to their therapeutic use. Authors should address all these issues also taking into account that irradiated MSCs undergo senescence (see for example PMID: 27288264).
We acknowledged this observation and consequently, we have modified the text according the reviewer suggestion:
We are aware that cellular therapy with MSCs can be problematic in cancer therapy (12). Therefore, it is important to emphasize that following irradiation MSCs become senescent and the senescence is associated with production of a senescent associated secretory phenotype (SASP). The SASP has anti-tumour activity since it may induce senescence of neighbouring cells by paracrine action (34). Nevertheless, SASP can modify its composition and become more and more rich in pro-inflammatory factors and it has become evident that tumour-associated MSCs have a positive effect on tumour growth and the spread of metastasis (35) through the acquisition of a chemo- and radio-therapy resistance mechanisms (36) and it has been suggested that tumour cells can misuse SASP for their own growth (37).
Lines 100-109 in the new manuscript
On the other hand, it has been communicated that, in an inflammatory condition the exosomes contained in the cancer cell secretome might have a role in the change of the normal MSC cell phenotype toward a malignant one (4) what could be an impediment to the MSC therapeutic use.
Lines 103-110 in the new manuscript
Thanks for your kind and constructive review that without doubt has improve our manuscript
Reviewer 2 Report
This review by Tovar Martín and colleagues discusses the rationale of using radiation activated mesenchymal stem cells (MSCs) for treating the acute respiratory distress syndrome. Based on recent observations published by the group, the review debates the idea of using exosomes released by the activated (irradiated with 2 Gy) MSCs, supposedly enriched in annexin-1, as therapeutic for acute respiratory distress syndrome and potentially for COVID-19 pneumonia. While the observation is interesting and may provide a therapeutic approach for ameliorating the symptoms of these diseases, I have several concerns as detailed below:
Major concerns:
- Figure 1: MSCs exert paracrine and exocrine actions, regardless of irradiation (activation) or not. Please, revise.
- The evidence of annexin-1 mRNA enrichment in the exosome fraction released by the activated MSCs is minimal. The claim that “We have described that there are qualitative, quantitative and functional differences amongst the proteins contained in the exosomes obtained from basal MSCs and activated MSCs* (39)” is not supported by the data presented in the published paper.
- Even if annexin-1 is well known for its beneficial effects in controlling the immune response, ageing, in maintaining homeostasis, etc., more work is needed for promoting the idea of using irradiated MSCs and their exosomes as a therapy in patients with ARDS; the specificity of the effects is not well demonstrated;
- The review is poorly organized; better analyzing and synthesizing the information/published literature presented to avoid the reiterations would significantly help.
Minor concerns:
- Brief discussion of engraftment efficiency and the fate of MSCs will significantly help the impact of this review.
- Potential side effects of stem cell therapy should be briefly mentioned;
- The review requires significant revision to avoid the numerous spelling errors, subject - verb agreement, lack of consistency for using the abbreviations, repeated words and even sentences, extremely long sentences/phrases.
- COVID-19 cannot invade the cells... COVID-19 is the disease, not the virus; please, revise.
Author Response
Reviewer 2
This review by Tovar Martín and colleagues discusses the rationale of using radiation activated mesenchymal stem cells (MSCs) for treating the acute respiratory distress syndrome. Based on recent observations published by the group, the review debates the idea of using exosomes released by the activated (irradiated with 2 Gy) MSCs, supposedly enriched in annexin-1, as therapeutic for acute respiratory distress syndrome and potentially for COVID-19 pneumonia. While the observation is interesting and may provide a therapeutic approach for ameliorating the symptoms of these diseases, I have several concerns as detailed below:
Major concerns:
- Figure 1: MSCs exert paracrine and exocrine actions, regardless of irradiation (activation) or not. Please, revise.
Done (Page 5 in the new manuscript)
- The evidence of annexin-1 mRNA enrichment in the exosome fraction released by the activated MSCs is minimal. The claim that “We have described that there are qualitative, quantitative and functional differences amongst the proteins contained in the exosomes obtained from basal MSCs and activated MSCs* (39)” is not supported by the data presented in the published paper.
We will try to clarified this observation here, modifying the text in some parts to facilitated its compression.
When we studied the exosome cargo before and after the activation of MSCs with RT, we discovered significant disparities in the results of the proteomic assessment of both samples. We have described that there are qualitative, quantitative and functional differences amongst the proteins contained in the exosomes obtained from basal MSCs and activated MSCs* (8). For more information see (8) and (8) supplementary materials, additional file 1: Significant biological process terms from REVIGO (Reduce + visualize gene ontology).
Lines 157-162 in the new manuscript
These findings demonstrate the profound metabolic change that these activated cell exosomes have undergone and the consequences after activation with radiation. Amongst the proteins representatives in exosomes released from MSCs* we highlight the key components of cell-cell or cell-matrix adhesion and include annexin and integrins (8). Between them, the presence of annexin A1 (ANXA1) is very noteworthy because it is always present in the exosomes released from MSCs* and constantly absent in MSCs. We have verified these results using quantitative mRNA-PCR to measure the mRNA of this protein in MSCs and MSCs* and confirmed that mRNA is spectacularly induced in MSCs after irradiation (8). After measuring quantitatively the mRNAs of the proteins of TRAIL, Dkk3 and ANXA1 in umbilical-cord stromal stem-cells, before and after cell stimulation with 2 Gy low-LET ionizing radiation, our results show a clear increase in their intracellular levels, compared with the levels found in basal situations (see these results in (8), supplementary material, Figure 2) and notice that the levels of mRNA of TRAIL and Dkk3 at 48 are strongly increased in treated cells compared to the basal levels (P<0.001), whereas the levels of mRNA of ANXA1 are strongly increased at 24 hours, and dramatically at 48 hours of cell treatment, with the statistical differences found 24 and 48 hours being very significant (P<0.0001), which supports the massive presence of ANXA1 in the exosomes released by the radiation-stimulated MSCs.
Lines 163-180 in the new manuscript
- Even if annexin-1 is well known for its beneficial effects in controlling the immune response, ageing, in maintaining homeostasis, etc., more work is needed for promoting the idea of using irradiated MSCs and their exosomes as a therapy in patients with ARDS; the specificity of the effects is not well demonstrated;
We agree with the referee and we have included the following sentence at the end of our paper
Finally, we want to clarify that this paper only presents a hypothesis and that the possibility of treating patients is still far away because we lack the necessary experimental data that would prove the applicability, efficiency and security necessary to further the hypothesis in its transition from the laboratory bench to the patient’s bed. Therefore, more work is necessary to promote this idea and use activated MSC as a therapy for patients with COVID-19, but that is our challenge and we are optimistic of a positive outcome.
Lines 466-471 in the new manuscript
.
- The review is poorly organized; better analyzing and synthesizing the information/published literature presented to avoid the reiterations would significantly help.
We have worked intensely to improve our paper and to respond to the constructive observations made by this reviewer.
Minor concerns:
- Brief discussion of engraftment efficiency and the fate of MSCs will significantly help the impact of this review.
According with your suggestion we have added the following paragraphs.
It is generally accepted that MSC based therapies are of major importance in regenerative medicine and, perhaps, in the future a solution for many other medical problems. However, the success of MSC therapy relies on the efficiency of its administration and the biodistribution, engraftment, differentiation, and secreting paracrine factors at the target sites (41). Until now, there has been no universal delivery route for mesenchymal stem cells (MSCs) for different diseases (42).
Lines 114-118 in the new manuscript
In fact, efficient homing and migration toward lesion sites play an important role and the local transplantation of MSC in spatial proximity to the lesion, as well as the systemic administration routes are being carefully explored (43). There is growing evidence that mesenchymal stem-cell based immuno-suppression was mainly attributed to the effects of MSC-derived extracellular vesicles (44) although it seems clear that transplanted MSC can indeed leave the blood flow and transmigrate through the endothelial barrier, and reach the lesion site (45). So, both mechanisms must be accepted until the underlying processes are better understood.
Lines 126-131 in the new manuscript
- Potential side effects of stem cell therapy should be briefly mentioned;
Nevertheless, whether this innate tropism of MSCs toward the tumours and metastatic foci is related with cancer promotion or suppression remains controversial (35, 37-40), and further studies on the interactions between cancerous cells and stromal components of tumour micro-environments have been proposed which is imperative to allow the progress of more suitable treatments for cancer.
Lines 114-118 I n tne new manuscript
It is generally accepted that MSC based therapies are of major importance in regenerative medicine and, perhaps, in the future a solution for many other medical problems. However, the success of MSC therapy relies on the efficiency of its administration and the biodistribution, engraftment, differentiation, and secreting paracrine factors at the target sites (41). Until now, there has been no universal delivery route for mesenchymal stem cells (MSCs) for different diseases (42). In fact, efficient homing and migration toward lesion sites play an important role and the local transplantation of MSC in spatial proximity to the lesion, as well as the systemic administration routes are being carefully explored (43). There is growing evidence that mesenchymal stem-cell based immuno-suppression was mainly attributed to the effects of MSC-derived extracellular vesicles (44) although it seems clear that transplanted MSC can indeed leave the blood flow and transmigrate through the endothelial barrier, and reach the lesion site (45). So, both mechanisms must be accepted until the underlying processes are better understood.
Lines 118-131 in the manuscript in the new manuscript
- The review requires significant revision to avoid the numerous spelling errors, subject - verb agreement, lack of consistency for using the abbreviations, repeated words and even sentences, extremely long sentences/phrases.
We have corrected all the manuscript and we believed that now is correct to be published.
- COVID-19 cannot invade the cells... COVID-19 is the disease, not the virus; please,
revise.
We have corrected this mistake.
Thanks for your review that without doubt has improve our manuscript
Round 2
Reviewer 1 Report
A minor comment: the authors have to use the same term also in the title.
They should write mesenchymal stromal/stem cells.
Author Response
Dear reviewer:
We have corrected all the small errors that you have indicated to us. Thank you very much for your careful review of our work. We believe that it has certainly improved it substantially.
Reviewer 2 Report
The revised version is improved. The only concern relates to grammar issues. Just some examples –
Line 124 – All of the above leads
Line 205 – we know that ANXA1 co-localize and exactly connect
Line 211 – Incomplete sentence
Line 285 – disfunction
Line 331 – clinical data suggests
Line 398 – The activation…produce
Abbreviations are not used consistently: A acute lung injury, annexin 1, mesenchymal stem cells
Author Response

(The authors gave the same response as above.)
